

# Rankings of extreme and widespread dry and wet events in the Iberian Peninsula between 1901-2016

Margarida L. R. Liberato[1,2], Irene Montero[1], Célia Gouveia[2,3], Ana Russo[2], Alexandre M. Ramos[2], Ricardo M. Trigo[2]

[1]Escola de Ciências e Tecnologia, Universidade de Trás-os-Montes e Alto Douro, UTAD, 5000-801 Vila Real, Portugal
[2]Instituto Dom Luiz, Faculdade de Ciências, Universidade de Lisboa, 1749-016 Lisboa, Portugal
[3]Instituto Português do Mar e da Atmosfera Lisboa, Portugal

*Correspondence to*: Margarida L. R. Liberato (mlr@utad.pt)

**Abstract.** Extensive, longstanding dry and wet episodes are one of the most frequent climatic extreme events in the Iberian Peninsula. Here, a method for ranking regional extremes of persistent, widespread drought and wet events is presented, considering different time scales. The method is based on the multiscalar Standardized Precipitation Evapotranspiration Index (SPEI) gridded dataset for the Iberian Peninsula. The Climatic Research Unit (CRU) data are used to compute SPEI between 1901 and 2016 by means of a log-logistic probability distribution function. The Potential Evapotranspiration (PET) is computed through the Penmann-Monteith equation. The ranking classification method is based on the assessment of the magnitude of an event, obtained after considering both the area affected respectively by the dryness or wetness – defined by SPEI values over a certain threshold – and its intensity in each grid point. A sensitivity analysis on the impact of different thresholds to define dry and wet events is performed. A comprehensive dataset of rankings of the most extreme, prolonged, widespread dry and wet periods in the Iberian Peninsula is presented, for aggregated time scales of 6, 12, 18 and 24 months. Results show that in the Iberian Peninsula there is not a region more prone to the occurrence of any of these long-term (dry and/or wet) most extreme events.

## 1 Introduction

Similarly to other Mediterranean regions, droughts represent one of the most frequent damaging natural disasters in the Iberian Peninsula (Sousa et al., 2011; Lionello et al., 2012; Trigo et al., 2013). In fact, drought events in Iberian Peninsula (IP) can impinge large socio-economic costs, often with widespread negative ecosystem impacts, such as significant losses in agricultural productions (e.g. Gouveia et al. 2009), hydro-electric production (e.g. Jerez et al., 2013) or increasing the risk of forest fires (e.g. Amraoui et al. 2013; Liberato et al., 2017). On the other hand, persistent large-scale precipitation periods may also be disruptive, being often responsible for high negative socio-economic impacts such as major floods, landslides, extensive property damage and even loss of human lives as described in the literature for recent wet winters occurring on IP (e.g. Zêzere et al., 2005; Vicente-Serrano et al., 2011; Sousa and Bastos, 2013). Drought is a complex phenomenon and differs from other natural hazards in various ways (Wilhite 1993). Its complexity is linked with the quantification of drought severity, which is usually performed by means of the drought impacts in different sectors ranging from economy, ecology, forestry and agriculture. The



identification of drought onset and end, and the quantification of its spatial extent and intensity represents
a challenge. The intricacy of this phenomenon is associated with the impossibility of finding a single
variable to measure to quantify the distinct characteristics of droughts (Vicente-Serrano et al., 2012).
Drought is essentially one consequence of an anomalous decrease of precipitation (Palmer, 1965), however
drought intensity varies both with the time scale (McKee et al., 1993) and spatial extend (Vicente-Serrano,
2006a). The World Meteorological Organization (WMO, 2006) presents the commonly accepted
classification of droughts in four categories. The meteorological drought is usually defined as a
precipitation departure from normal over a predefined period of time. The agricultural drought is usually
defined in terms of needed soil water during the growing season to support healthy crop growth up to;
depends of precipitation rate but also of soil characteristics that favor its water-holding capacity. The
hydrological drought usually results from the deficiencies in surface and subsurface water supplies, causing
in the drying up of reservoirs, lakes, rivers and decline in groundwater level; it depends on multiple and
competing factors such as irrigation, hydroelectric power, tourism and ecosystem management among
others. There is also a considerable time lag between precipitation events and their impacts on surface and
subsurface components of the hydrologic system, being the recovery also slow due to the long recharge
periods. Finally, the socio-economic drought, associating droughts with supply and demand of an economic
good that is dependent on precipitation.
Drought analysis and monitoring have been conducted using several methodological approaches. The
demanding task of objectively identifying the onset and end of a drought together with the quantification
of drought severity led to the development, in recent years, of a new set of drought indicators (Vicente-
Serrano et al., 2010a), namely the Palmer Drought Severity Index (PDSI, Palmer, 1965), founded on a soil
water balance equation; the Standardised Precipitation Index (SPI; McKee et al., 1993) built using a
precipitation probabilistic approach; or the Standardized Precipitation Evapotranspiration Index (SPEI),
calculated using precipitation and temperature fields together. The main advantage of SPEI is combining
the multi-scalar character of SPI with the ability of including the effects of temperature variability on
drought assessment (Vicente-Serrano et al., 2010b). One of the most used drought indices worldwide is the
PDSI (Palmer, 1965), however this index has been shown to provide unreliable results in Europe, presenting
exaggerated frequency of extreme dry or wet spells (e.g. van der Schrier et al., 2006; Sousa et al., 2011).
Thus, several authors have made an effort to adapt it and developed the regionally more consistent self-
calibrated PDSI (scPDSI) with better results over the Mediterranean basin (Sousa et al., 2011).
Nevertheless, this index does not allow for a multi-scale approach.
Several studies show the relationship between the temporal variability of drought indices and the response
of natural ecosystems, such as tree growth (De Soto et al., 2014), river discharge (Vicente-Serrano and
López-Moreno 2005), crop yields (Vicente-Serrano et al., 2006; Páscoa et al., 2017) and vegetation activity
(Vicente-Serrano et al., 2013; Liberato et al., 2017). The multi-scalar character of SPI and SPEI indices
enables to quantify different types of droughts, something that neither PDSI nor the enhanced scPDSI can
provide. The shorter times scales (between 3 and 6 months) can be used to describe agricultural droughts
(McKee et al., 1993) as they allow to monitor the moisture conditions relative to soil and vegetation
(Vicente-Serrano, 2006). The medium length time scales (between 6 and 12 months) are often useful to
evaluate hydrological droughts due to their ability for monitor surface water resources (Vicente-Serrano



and López-Moreno, 2005). Longer time scales (24 to 36 months) indicate longer nonetheless less frequent
droughts, with fewer wet or dry periods (Vicente-Serrano, 2006).
Vicente-Serrano and co-authors (2012) compared the performance of the above-mentioned drought indices
and found that both SPEI and SPI, both computed for different time scales, have improved ability to capture
drought impacts on hydrological, agricultural and ecological systems. Additionally, SPEI is the index that
best caught the responses of the considered meteorological variables to drought in summer and are also
more sensitive to global warming (Vicente-Serrano et al., 2010b), as the general (not for a specific species)
water balance is included in the computation of SPEI through the calculation of the difference between
monthly precipitation and the potential evapotranspiration. Moreover, as PDSI, SPEI and SPI allow the
quantification of both wetness (positive value) and dryness (negative values). SPI and SPEI are also
standardized variables allowing a detailed analyses of droughts across sites with very different climatology.
In the last years, both have been used to characterize the dry and wet periods in several regions, namely the
SPI for United States (Wu et al., 2007), Italy (Vergni and Todisco 2010) and China (Du et al, 2013) and
SPEI for Czech Republic (Potopová et al., 2015) and in China (Tao et al., 2015) and Central Europe
(Spinoni et al., 2013).
The characterization of wet and dry periods in the IP becomes extremely important as the region is
frequently affected by extreme dry and wet events and consequently by their impacts on several systems,
with significant damages which justify the usage of several time scales in the analyses, ranging from 6 to
24 months. Thus, it is important to make the assessment of the extreme wet and dry periods affecting an
extensive area of IP at longer time scales. Therefore, this work has the main goal of building rankings of
the most severe, widespread dry and wet periods on Iberia, at several time scales, as well as analyzing their
evolution on time since the beginning of the 20th century to present.
In summary the main goals of this paper are to:
(i) present a tool which allows identifying regional extremes of persistent, widespread dry and wet periods,
at different time scales;
(ii) build a comprehensive dataset of rankings of the most extreme, prolonged, widespread drought and wet
periods on Iberia, for time scales 6, 12, 18 and 24, spanning the period from 1901 to 2016, using the multi-
scalar SPEI gridded dataset with a regular resolution of 0.5 degree.
This paper is organized as follows. Section 2 contains a description of the data and methods used, including
a brief overview of the SPEI index regionally computed for Iberia. In section 3 results are presented for
drought rankings obtained for different time scales while results for the wet periods are discussed in section
4. Finally, discussion and conclusions are presented in section 5.
**2 Data and Methodology**
**2.1 SPEI datasets**
The Climatic Research Unit (CRU) TS4.01 high resolution gridded data for the period 1901-2016 are used
in this study to obtain the SPEI timeseries. The CRU TS4.01 dataset includes month-by-month time-series
which are calculated on high-resolution (0.5x0.5 degree) grids based on an archive of monthly average





daily data provided by more than 4000 weather stations distributed around the world (Jones and Harris,
2013). At this resolution the study region of IP corresponds to a square of 30x30 grid pixels.
The CRU TS43.01 variables include cloud cover, diurnal temperature range, frost day frequency, potential
evapotranspiration (PET), precipitation, daily mean temperature, monthly average daily maximum and
minimum temperature, vapor pressure and wet day frequency. Thus, this dataset is very suitable for the
study of climate variability and also to drought analysis. The CRU Potential Evapotranspiration (PET) used
is based on the Penmann-Monteith equation. This method is considered as the standard procedure for
computing PET by several international institutions such as the Food and Agriculture Organization of the
United Nations (FAO), the International Commission on Irrigation and Drainage (ICID), or the American
Society of Civil Engineers (ASCE). The log-logistic probability distribution is also used to fit SPEI and the
L-moment method is used for the parameters estimation. This formulation allows a very good fit to the
series of differences between precipitation and PET (Vicente-Serrano et al., 2010b).
Values of the SPEI range mostly between -2.5 and 2.5, corresponding to exceedance probabilities of
approximately 0.006 and 0.994, respectively, although the theoretical limits are (-∞, ∞). The severe dry and
wet events were selected based on a SPEI threshold of -1.28 and 1.28 respectively that correspond
approximately to 10% of the extreme cases according with the probability distribution function (Agnew,

131 2000).

The use of a long-term period (116 years) allows for the identification of spatial patterns of droughts over
the IP, increasing the knowledge on the most intense and wide-ranging droughts in the IP and their temporal
variability on a climatic perspective, while classifying the extreme events and the drought-prone areas.
The usage of datasets which include data prior to 1950 is widely discussed (e.g. Sousa et al., 2011).
Nevertheless, it should be accounted that the CRU TS 4.01 dataset originates from thousands of stations
dispersed nonrandomly, with higher densities at mid-latitudes (Macias-Fauria et al., 2014). As the analysis
of the present work is constrained to the IP, which includes a relatively homogeneous number of stations,
we consider that the quality of the data ensures a robust analysis, keeping in mind that the number of stations
used has varied over time. Supplementary information on stations' availability and on the interpolation
methods used is provided in New et al. (2000) and Mitchell and Jones (2005). Regardless of the smaller
number of meteorological stations available until 1950's comparatively to the remaining period, Harris and
co-authors (2013) have shown that precipitation and temperature time series are highly correlated with other
datasets (Harris et al., 2013). Moreover, this database has been formerly used by the authors (Russo et al.,
2015; Páscoa et al., 2017a), which have attained good results in the IP, including for the earlier years
(Páscoa et al., 2017a,b).
**2.2 Ranking extreme widespread drought and wet events**
As aforementioned one of the main aims of this study is to characterize and rank the most extreme,
widespread drought (wet) events. Thus, only the severe drought (wet) events should be selected based on a
SPEI threshold of -1.28(+1.28) that corresponds to the 10% of the extreme cases according with the
probability distribution function. This procedure allows selecting, for each grid point, a dataset of extreme
drought (wet) events which is then analyzed. Following the approach used for daily extreme precipitation





in Iberia (Ramos et al., 2014), here for each month of the dataset, the spatial extension of the events is
assessed by multiplying, for each month and time scale:
- the mean value of the SPEI for all the grid points selected from the threshold criteria above – i.e.,
below(above) the threshold of -1.28 (+1.28);
- by the area (hereafter A, in percentage) affected by that extreme value.
It is worth mentioning that the sensitivity of this ranking to the chosen threshold has also been assessed.
Several tests have been performed, namely by using a less restrictive value (-0.83 and +0.83 respectively
for moderate dry and wet events). As expected, these changes imply that the absolute mean values are
reduced while the area increases. Final results do not change significantly the top rank of the most extreme
events, even though some years appear in different rank order.
This methodology may be applied to all months of all time scales and to different domains. This would
implicate a very large number (288=12months x 24 time scales) of different rankings for each domain. For
the sake of clearness and in view of the physical interpretation of extreme events occurring on the IP (Figure
1), the analysis is performed for specific months for each of the 4 time scales (6, 12, 18 and 24 months)
resulting from the following reasoning. For shorter time scales, the 6-month time scale, which allows
describing the agricultural droughts (McKee et al., 1993), it is assessed for the month of March. This will
represent the winter (October to March) drought or wet period over the IP. The 12-month time scale is
represented by the values obtained for the month of September, therefore englobing the hydrological year,
particularly suitable for monitoring surface water resources (Vicente-Serrano and López-Moreno, 2005).
Additionally, the longer time scales – 18-month for March and 24-month for September – permit to study
longer multi-annual but less frequent droughts.
The rank index |R| is used for ranking the extensive dry events for the IP, taking into account not only the
severity of the extended winter (October to March) drought but also its spatial extension which may be
evaluated by the affected area (A, in %) of the IP that has SPEI values surpassing the chosen threshold (in
this case only the 10% most severe and extreme events, characterized by SPEI < -1.28 at each grid point).
The mean SPEI value which characterizes the drought event is omitted in the table for the sake of simplicity,
since it may be easily obtained from the quotient between |R| and A (being negative (positive) for dry (wet)
events).
**3 Extensive extreme droughts during the period 1901-2016**
The methodology described in the previous section has been applied successively to the several datasets,
time scales and domains considered, thus generating several different ranking lists. Here the SPEI dataset
obtained from the CRU TS4.01 high resolution (0.5x0.5 degree) gridded data for the period 1901-2016 has
been applied at four time scales for the IP domain, with the main purpose to identify the major extensive,
extreme droughts which affected the IP while illustrating the relevance of the method. In this section the
ten most extreme events (Top #10) for each time scale are presented in Table 1, the six major droughts (Top
#6) are shown in Figure 2, and some examples are discussed in detail.



### 3.1 Extreme agricultural droughts (6-month time scale)

Table 1 shows the Top #10 absolute final rank index |R| for the 6-month time scale obtained for March. As expected, the Top #10 episodes identified in Table 1, for the 6-month time scale (March) correspond to well-known droughts which had high negative impacts on the IP. The Top #1 (mean SPEI06 of -2.00) is the well-known drought episode of the 2011-2012 winter (Trigo et al., 2013). The extensive spatial extent of these droughts is clearly illustrated in the 1st column of Figure 2, which shows the grid points which, in March of the corresponding year, had SPEI values lower than -1.28 (at the 6-month time scale). Another important aspect resulting from these results is that the most extreme episodes correspond also to those episodes which affected a larger area. In fact the top 6 events represented in the 1st column of Figure 2 had more than 50% of the IP area on severe or extreme drought. Additionally, from the 1st column of Figure 2, there is no evidence of the existence of a particular region on the IP which might be more prone to extreme or severe agricultural droughts, even though there is prevalence on the western sectors of the IP, specially affecting Portugal.

### 3.2 Extreme hydrological droughts (12-month time scale)

As above-mentioned, the 12-month time scale SPEI for the month of September is here used to assess severe and extreme hydrological (October of year n-1 to September of year n) droughts, which represent important negative impacts on the water resources. The analysis of Table 1 shows that most of the Top #10 episodes of agricultural droughts (SPEI06) developed into extreme hydrological droughts (SPEI12) which are also in the respective Top #10. In fact, the 2004-2005 (Top #15) drought event (García-Herrera et al., 2007) which is here classified as the fifteenth most extreme agricultural drought in IP (mean SPEI06 of -1.48 over circa 32% of Iberia), is now the Top #1 (mean SPEI12 of -2.17) on 12-month time scale, with more than 73% of the IP on severe or extreme drought (2nd column of Figure 2); the 2012 (SPEI06 Top #1) drought is now the Top #2 (mean SPEI12 of -1.73) on 12-month time scale, with more than 76% of the IP on severe or extreme drought (Figure 2); the 1945 (SPEI06 Top #7) drought is now the Top #6 (mean SPEI12 of -1.82) on 12-month time scale, with almost 62% of the IP on severe or extreme drought (Figure 2). Naturally, some events can drop even more significantly in the rank, thus the 2000 (Top #2) drought is now the Top #19 (mean SPEI12 of -1.59) on 12-month time scale, with only 17% of the IP on severe or extreme drought. From Table 1 it is noticeable that the spatial extent of the most extreme hydrological droughts is in general larger. The area of 2012 (Top #1 SPEI06) is 74%, while its area (Top #2 SPEI12) increases to 76%; and the area of 1945 (Top #7 SPEI06) increases from 49% to 62%. On the contrary the disappearance of some events from the Top 10 is generally due to a reduction on the affected area. This decrease in the affected area is evident from the inspection of the 2nd column of Figure 2.

### 3.3 Extreme persistent droughts (18 and 24-month time scales)

The same methodology has also been applied to study longer, persistent extreme droughts. As above-mentioned, the 18-month time scale SPEI for the month of March (3rd column of Figure 2) and the 24-month time scale SPEI for the month of September (4th column of Figure 2) are used to assess severe and extreme persistent droughts, which represent major negative societal impacts. The analysis of Table 1 shows that on the IP most of the Top #10 episodes of extensive agricultural and hydrological droughts





developed into extreme persistent droughts which are also in the Top #10 on 18 and 24-month time scales.
Examples are the 2004-2005 and 2011-2012 droughts, alternating on Top #1 and Top#2 on both time scales
(2005: A18=72%; A24=80%) – 2012: A18=79%; A24=80%), the 2015-2016 (Top #9, A18=33% and Top
#6, A24=53%) and the 1995 (Top #7, A18=39% and Top #5, A24=64%) droughts.
In summary, when applying this method developed to take into consideration the two factors, namely the
area of influence of the drought (given by SPEI values above a certain threshold) and the severity of the
episode (given by the mean values of the SPEI above the chosen threshold), the well-known widespread,
extreme droughts occurring in the IP are correctly hierarchized, at the different time scales.
**3.4 Frequency of widespread droughts**
The method developed has been applied until now only to the severe and extreme droughts, by considering,
for each month, the -1.28 threshold (i.e., pixels with SPEI < -1.28). Since it is important to evaluate the
sensitivity of the results to this criterion, similar rankings have been built with the -0.83 threshold, for the
same time scales. Therefore, these new ranking lists correspond to widespread moderate, severe or extreme
droughts. As expected by relaxing the severity criterion (from -1.28 to -0.83) the mean values of SPEI
decrease, while the area affected by the droughts increases. Consequently, the years on the Top #10 ranking
lists remain mostly the same, although they may occupy a different ranking order. Exceptions are the Top
#9 for SPEI06, the Top #10 for SPEI18 and the Top #8 and Top #10 for SPEI24 which are no longer
classified on the 10 most extreme episodes. Unsurprisingly the SPEI06 ranking list, corresponding to
agricultural droughts, is the most sensitive to the threshold choice. The consistency among ranking lists is
evident on the analysis of the time evolution of the ranking index obtained from the SPEI as depicted though
all the different time scales considered (Figure 3). In this figure, the blue lines correspond to the ranking
indices obtained from only severe and extreme droughts (threshold -1.28) and the red lines represent the
ranking indices obtained from moderate to extreme droughts (threshold -0.83). The correlation for each pair
of lists is between 0.962 and 0.966.
Figures 3 and 4 summarize well the abovementioned results.
- Most widespread agricultural droughts have correspondence on longer time scales. Thus, the most extreme
extensive agricultural droughts evolve into hydrological and more persistent extreme droughts.
- There is a clear temporal clustering of most extreme drought episodes, particularly with a large
concentration between 1943 and 1957 and a second group after 1975. This is valid independently of the
considered time scales.
- The difference between the blue curve (moderate, severe or extreme) and red curve (only severe or
extreme episodes) is reduced during the most extreme drought episodes.
- Most moderate drought episodes are coincident with severe or extreme drought, even though it has usually
a smaller index – and thus a smaller extension, as confirmed in Figure 4.
- The frequency of extensive episodes is almost the same at all time scales considered – widespread
moderate drought episodes, even if they have smaller extent (Figure 4), occur at the 4 time scales analyzed.



### 4 Extensive extreme wet events during the period 1901-2016

Similar analysis is now performed for the widespread, moist periods occurring on IP during the 1901-2012 period. Table 2 shows the Top #10 absolute final rank index |R| used for ranking the extensive wet events on the IP, for the 4 considered time scales. Figure 5 shows the grid points which had SPEI values higher than +1.28 for the wet Top #6 events (at the 4 considered time scales). As previously stated, this index takes into account not only the severity of the wet episodes but also their spatial extension which may be evaluated by the affected area (A, in %) of the IP that has SPEI values surpassing the chosen threshold (in this case only the 10% most severe and extreme events, characterized by SPEI > +1.28 at each grid point). As on Table 1, the mean SPEI value which characterizes the wet event is omitted in Table 2 for the sake of simplicity, since it may be easily obtained from the quotient between |R| and A (being always positive for wet events).

The first noticeable result is that the method performs equally well when we build the ranking lists for extensive, extreme moist events, and therefore, analogous statements can be raised concerning the top rank tables. As expected, some of the events represented on the Top #10 correspond to well-known very moist periods on the IP, such as the 2010 winter (Vicente-Serrano et al., 2011; Liberato et al., 2013), the 2001 winter (Sousa et al., 2011; Ramos et al., 2014).

The consistency among ranking lists is also evident from the analysis of the time evolution of the ranking index obtained from the SPEI at all time scales considered (Figures 3 and 4). Grey lines correspond to the ranking indices obtained from only severe and extreme wet events (threshold +1.28) and the black lines represent the ranking indices obtained from moderate to extreme moist periods (threshold +0.83). The correlation between time series relative to each pair of lists is very high, ranging between 0.950 and 0.965. Additionally, a black dashed line is represented, corresponding to the ranking indices obtained from severe and extreme droughts (threshold -1.28; same as blue line on Figure 3). As expected the extensive droughts are anticorrelated to the extensive wet periods – values in the range between -0.243 to -0.297 (significant at the 1.08% level)

Figure 4 summarizes well the abovementioned results.

- Most widespread wet events have correspondence on longer time scales.

- Most extreme wet episodes occurred between 1936 and 1941 and between 1959 and 1979, in all time scales.

- In the most extreme cases the difference between moderate, severe or extreme and only severe or extreme episodes is reduced.

- Most widespread moderate wet episodes are anti correlated with severe or extreme drought, and they have usually a smaller index – and thus a smaller extension.

- The frequency of widespread episodes is almost the same at all time scales considered – widespread moderate moist episodes, even if they have smaller extent, occur at the 4 time scales analyzed.

### 5 Concluding remarks

Extreme dry and wet events are usually disruptive events and the associated impacts may differ considerably depending on the extension of the affected area. Therefore, a method for ranking regional



extremes of persistent, widespread drought and wet events is presented in this paper, considering different
time scales (6, 12, 18 and 24 months). The method is based on the multiscalar Standardized Precipitation
Evapotranspiration Index (SPEI) gridded dataset for the Iberian Peninsula for the period 1901-2016.
For both the dry and wet periods, this tool allows identifying well known regional extremes of persistent,
widespread dry and wet periods, at different time scales. Additionally, a comprehensive dataset of rankings
of the most extreme, prolonged, widespread drought and wet periods on Iberia, for time scales 6, 12, 18
and 24 months, spanning the period from 1901 to 2016, using the multi-scalar SPEI gridded dataset with a
regular resolution of 0.5 degree is built, and will be available upon request, for the 8 domains: Iberian
Peninsula, for Portugal and each of the six Iberian regions. Future work will now be performed in the frame
of current projects based on the identified most extreme events in order to better understand the common
mechanisms behind each of these events.
**Data availability.**
The rankings' datasets will become publicly available. For further information, please contact the
corresponding author.
**Author contributions.**
AR performed the SPEI calculations and provided the datasets, IM performed the ranking calculations,
MLRL, IM and AR designed and made the figures, MLRL and CG designed the study and wrote the article,
and all authors contributed to the interpretation, discussion of the results and revision of the manuscript.
**Competing interests.**
The authors declare that they have no conflict of interest.
**Acknowledgments**
This work was supported by national funds through FCT (Fundação para a Ciência e a Tecnologia, Portugal)
under project QSECA (PTDC/AAGGLO/4155/2012), project IMPECAF (PTDC/CTA-CLI/28902/2017),
and project UIDB/50019/2020 - IDL.

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

**Figures**

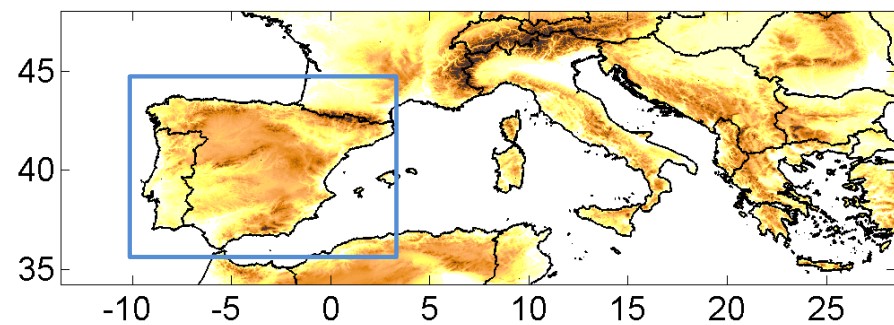


**Figure 1: The Iberian Peninsula (blue box) in the Mediterranean region.**






**Figure 2: Top six drought events in the Iberian Peninsula domain for: 1st column: 6-month time scale SPEI for March (2012; 2000; 1983; 1907; 1961 and 1992); 2nd column: 12-month time scale SPEI for September (2005; 2012; 2009; 2015; 2016; and 1945); 3rd column: 18-month time scale SPEI for March (2012; 2005; 1949; 2008; 1992; and 1981); 4th column: 24-month time scale SPEI for September (2005; 2012; 2015;1945; 1995; and 2016).**








**Figure 3: Time evolution of the (a) agricultural (ranking index obtained from March SPEI at 6-month time**
**scale), (b) hydrological (ranking index obtained from September SPEI at 12-month time scale) and (c-d) longer,**
**persistent (ranking indices obtained from March SPEI at 18-month time scale (c) and September at 24-month**
**time scale (d)) droughts. The blue lines correspond to the ranking indices obtained from severe and extreme**
**droughts (threshold -1.28); the red lines represent the ranking indices obtained from the most extreme droughts**
**(threshold -1.65). The black lines represent analogous time evolution of the moist ranking index obtained for**
**each timescales (threshold 1.28 (black) and 1.65 (grey)).**





Figure 4: Similar as Figure 3 but for the area (A, in percentage), for the 4 timescales.



**Figure 5: Top six wet events in the Iberian Peninsula domain for: 1st column: 6-month time scale SPEI for March; 2nd column: 12-month time scale SPEI for September; 3rd column: 18-month time scale SPEI for March; 4th column: 24-month time scale SPEI for September.**





481

**Tables**

**Table 1** – Ten most dry extreme events (Top #10) for each time scale. Absolute final rank index |**R**| for the 6-month time scale obtained for March (SPEI 06); 12-month time scale obtained for September (SPEI 12); 18-month time scale obtained for March (SPEI 18); 24-month time scale obtained for September (SPEI 24). A represents the area (in percentage) where the respective index in below the -1.28 threshold.

| Rank No. | SPEI 06 (March) | | | SPEI 12 (September) | | | SPEI 18 (March) | | | SPEI 24 (September) | | |
|---|---|---|---|---|---|---|---|---|---|---|---|---|
| | Year | A (%) | \|R\| | Year | A (%) | \|R\| | Year | A (%) | \|R\| | Year | A (%) | \|R\| |
| 1 | 2012 | 73.86 | 147.47 | 2005 | 73.30 | 158.96 | 2012 | 78.98 | 172.34 | 2005 | 79.83 | 177.06 |
| 2 | 2000 | 69.32 | 129.45 | 2012 | 76.14 | 131.91 | 2005 | 71.59 | 114.41 | 2012 | 79.55 | 166.70 |
| 3 | 1983 | 56.25 | 91.50 | 2009 | 68.47 | 126.58 | 1949 | 62.22 | 109.02 | 2015 | 67.90 | 116.68 |
| 4 | 1907 | 54.55 | 87.60 | 2015 | 67.33 | 122.75 | 2008 | 62.50 | 100.84 | 1945 | 61.65 | 115.75 |
| 5 | 1961 | 55.97 | 85.96 | 2016 | 60.51 | 114.43 | 1992 | 53.13 | 99.07 | 1995 | 63.64 | 109.46 |
| 6 | 1992 | 51.14 | 85.96 | 1945 | 61.65 | 112.38 | 1981 | 49.43 | 78.94 | 2016 | 52.56 | 96.22 |
| 7 | 1945 | 49.15 | 75.89 | 2011 | 63.35 | 108.24 | 1995 | 39.20 | 72.32 | 2009 | 49.72 | 86.08 |
| 8 | 1993 | 44.89 | 72.21 | 2003 | 53.13 | 90.60 | 2000 | 40.06 | 61.41 | 2011 | 37.22 | 62.68 |
| 9 | 1995 | 40.63 | 71.10 | 1990 | 53.69 | 85.52 | 2016 | 32.95 | 59.81 | 2014 | 35.80 | 58.90 |
| 10 | 1944 | 44.60 | 68.71 | 1995 | 39.77 | 59.18 | 1999 | 38.64 | 57.09 | 2006 | 34.94 | 50.10 |

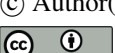


**Table 2** – Ten most wet extreme events (Top #10) for each time scale. Absolute final rank index |R| for the 6-month time scale obtained for March (SPEI 06); 12-month time scale obtained for September (SPEI 12); 18-month time scale obtained for March (SPEI 18); 24-month time scale obtained for September (SPEI 24). A represents the area (in percentage) where the respective index in above the 1.28 threshold.

| Rank No. | SPEI 06 (March) | | | SPEI 12 (September) | | | SPEI 18 (March) | | | SPEI 24 (September) | | |
|---|---|---|---|---|---|---|---|---|---|---|---|---|
| | Year | A (%) | \|R\| | Year | A (%) | \|R\| | Year | A (%) | \|R\| | Year | A (%) | \|R\| |
| 1 | 1936 | 75.57 | 158.31 | 1936 | 77.56 | 148.42 | 1936 | 70.74 | 146.13 | 1936 | 80.11 | 188.77 |
| 2 | 1947 | 67.05 | 133.47 | 1969 | 70.45 | 130.97 | 1960 | 68.18 | 130.72 | 1969 | 69.89 | 126.87 |
| 3 | 1969 | 66.48 | 115.28 | 1971 | 68.75 | 125.54 | 1964 | 47.16 | 73.84 | 1941 | 67.90 | 121.97 |
| 4 | 1979 | 61.36 | 105.84 | 1959 | 53.69 | 105.41 | 1969 | 45.45 | 71.95 | 1977 | 50.85 | 83.91 |
| 5 | 1960 | 58.52 | 91.95 | 1941 | 52.27 | 85.30 | 1947 | 42.33 | 69.82 | 1959 | 39.77 | 71.10 |
| 6 | 2010 | 46.02 | 91.31 | 1972 | 42.05 | 77.59 | 2001 | 36.08 | 66.47 | 1963 | 40.63 | 70.86 |
| 7 | 2013 | 49.43 | 77.62 | 1963 | 42.05 | 70.42 | 2010 | 34.66 | 64.19 | 1956 | 43.18 | 69.29 |
| 8 | 1919 | 38.07 | 63.48 | 1977 | 42.05 | 67.90 | 1941 | 38.64 | 63.49 | 1972 | 34.38 | 66.91 |
| 9 | 1956 | 41.76 | 63.18 | 1932 | 39.20 | 66.83 | 1962 | 39.20 | 62.03 | 1912 | 31.53 | 51.05 |
| 10 | 1941 | 36.93 | 54.10 | 1939 | 44.03 | 65.90 | 1979 | 36.08 | 58.47 | 1960 | 31.25 | 46.89 |

482