# Peer review of "Rankings of extreme and widespread dry and wet events in the Iberian Peninsula between 1901-2016"

_Earth System Dynamics, 2020_

## Short Comment (SC1) · 24 Aug 2020

1) Line 26-27 -> the term "natural disasters" is now obsolete (see e.g. [1] https://www.gndr.org/news/item/1499-disasters-are-not-natural.html and/or [2] https://www.undp.org/content/undp/en/home/ourperspective/ourperspectivearticles/2012/10/12/what-we-call-natural-disasters-are-not-natural-at-all-jo-scheuer/). I suggest to rephrase the word to "natural hazards".

2) If you find it useful you can cite the following work where a global analysis of concurrent flood-drought events has been performed using the PDSI https://doi.org/10.5194/esd-11-251-2020

[Figure]

Best, paolo

---

## Referee Comment (RC1) · Anonymous Referee #1 · 16 Sep 2020

This study describes the main dry and humid events that have affected the Iberian Peninsula from 1901 to 2016. The research topic addressed is relevant neverthe-less, the study is principally descriptive. This is not a limitation per se, but further discussion should be necessary to justify the relevance and novelty of this study in comparison to the several studies (some of them very recent) analysing droughts from different perspectives in the Iberian Peninsula. In addition, there are some method-ological issues that should be considered to remove some interpretations of the ob-tained results. There are also some data issues and recommendations to improve the manuscript. I would suggest a major revision of this manuscript. Comments: 42. typo: Spatial extend 44. I would also mention environmental or ecologic droughts

(e.g. https://doi.org/10.1016/j.earscirev.2019.102953) 44-54. I do not think necessary here to describe in depth these drought types. 58. The PDSI is not recent. I suggest to cite a study that reviews drought indices (e.g. https://doi.org/10.1175/1520-0477-83.8.1149, https://doi.org/10.1007/s40641-018-0098-x). 63-68. I would remove this criticism of the PDSI. I do not it is needed here. 85. This is not a water balance model. The SPEI is not intended to be soil moisture metric. 93-99. I would suggest to revise recent literature on this topic also in the Iberian Peninsula (https://doi.org/10.1002/joc.6719, https://doi.org/10.1007/s11600-018-0138-x, https://doi.org/10.1111/nyas.14365, https://doi.org/10.1002/joc.6126). 115. Nevertheless, the relevant issue is the number of stations used in the Iberian Peninsula. This is very relevant in this study. This is very relevant in this study since this may affect the obtained results since few interpolated stations may filter too much the spatial variance of the specific drought episodes. 117. I wonder on the goodness of the VPD data for the first decades of the XXth century. 132-146. There are strong uncertainties related to the calculation of the evaporative demand prior 1960s. This explains that some drought datasets based on the AED start in the decade of 1960 (https://doi.org/10.3390/data2030022). I am not convinced by the arguments provided to justify the use of the CRU dataset for the long-term, mostly for the atmospheric demand, at least in Spain few of the data necessary to calculate the AED with the FAO-56 method is not available. If author's main focus is the long term, I would recommend focusing only on precipitation data and the SPI. This would reduce uncertainty since CRU precipitation data shows good capacity to reproduce long-term precipitation variability in the region (https://doi.org/10.1088/1748-9326/ab9c4f). 183-185. Repeated above. 166. I suggest restricting the analysis to time scales below 12 months. The autoregressive character of SPEI and SPI affect the identification of the drought events and the possible trends (e.g., https://doi.org/10.1002/joc.6350, https://doi.org/10.1175/JCLI-D-15-0590.1, https://doi.org/10.1016/j.scitotenv.2020.140094). Fig. 2 Most of the events identified are recorded in the decades of 2000 and 2005. I would ask if AED uncertainties may be biasing this issue. Droughts based on precipitation data do not show so important trends (https://doi.org/10.1002/joc.6719). Also SPEI data based on high qual-ity data in Spain does not show so important trend (https://doi.org/10.1002/joc.6126). 221-234. Remove 18 and 24 months. Persistence should be also recorded with 6- and 12-month SPEI and it is not affected by the mentioned autocorrelation problems. 236-244. I would not use different thresholds. This makes the structure of the article more complex unnecessarily. Figure 1. Is this needed? Discussion: There is not a discussion section that discusses in depth the obtained results and some comparison with the scientific literature related to droughts published in the Iberian Peninsula in re-cent years. I think extremely necessary to write an independent discussion section in which the caveats of the data and methodology are stressed, but also the implications in the current climate change scenario and the possible novelty of the results in relation with the existing literature.

---

## Referee Comment (RC2) · Anonymous Referee #2 · 21 Sep 2020

The main goal of this study is to characterize and rank the most extreme, widespread drought (wet) events in the Iberian Peninsula between 1901–2016. The method used is based on the multiscalar Standardized Precipitation Evapotranspiration Index (SPEI) from CRU. Besides, different time scales (6, 12, 18 and 24 months) were considered in ranking. The topic is quite interesting, but there is a lack of further discussion of the results. It is also necessary to include some validation of the most severe drought events identified in the study area.

General Comments:

Abstract: The authors could bring more highlights about the results in the abstract.

[Figure]

Introduction: Line 57: It is not correct to say that drought indices have been developed in recent years. The PDSI, for example, was created more than 50 years ago. Please modify the sentence. Lines 106-109: Perhaps these sentences can be removed. I do not think it is necessary to describe the topics of the article. The authors could add more details in the introduction about the reason for using SPEI instead other indices, pointing out the main advantages and disadvantages. Extensive extreme droughts during the period 1901-2016: How the major negative societal impacts related to severe droughts was measured? Please, add some examples in the manuscript. Please consider including some information on the possible causes (climate) of the most severe drought events in the study area. In addition, it is necessary to quantify the droughts impacts on agricultural production and reservoirs, as a validation of the ranking of drought. Figure 1 - What information is presented by the color scale in figure 1? Please specify if it is a physical variable (elevation, etc.).

---

## Referee Comment (RC3) · Anonymous Referee #3 · 26 Sep 2020

Main comments: The manuscript presents a new method for ranking regional extremes of persistent, widespread drought and wet events that considers different time scales. While there are some good points in the manuscript, it lacks in discussion where the authors could put things in context. Specific questions that need to be addressed are:

• Can you show that this method using SPEI performs better for the Iberian Peninsula? How does it compare with other indices (say SPI) for the same purpose? Can you validate its applicability for the outcomes (hydrological/ ecological/ agricultural) on different time-scales? It should be fairly easy to show like Vicente-Serrano et al., 2012 did (for global and continental scales) that SPEI is better than SPI (and PDSI) for vari-

ous applications. I wonder if it is true for the Iberian Peninsula also? • The question of ranking also needs more exploration. Does a top ranking translate into maximum impacts on the indicators (agriculture/ streamflow etc.) for the relevant time-scale? Can this be shown in an analysis? • The temporal variation in the indices (especially in Figures 3 & 4) described as "There is a clear temporal clustering of most extreme drought episodes, particularly with a large concentration between 1943 and 1957 and a second group after 1975". • Some analysis should be done (or connections to existing studies made) to analyze the role of modes of variability on decadal or multi-decadal or inter-annual (such as the NAO by Vicente-Serrano et al. (2011) time-scales. Are there any studies carried out over a larger region - like say the Mediterranean – that one can make a connection to? • Is the clustering of drought events towards the end of the data period in any way connected to climate change? There are numerous studies that have already documented drying of the Mediterranean under climate change.

Other suggestions: • Numerous grammatical mistakes that can be easily fixed in a modern word processor.

Please also note the supplement to this comment:
https://esd.copernicus.org/preprints/esd-2020-46/esd-2020-46-RC3-supplement.pdf

---

## Author Comment (AC1) · 2 Nov 2020

Reply: Thank you for your interest and for the constructive comments, which may contribute to improving the manuscript. We will incorporate your first suggestion in the text. The second will not be included as the paper does not intend to make a review on concurrent events and approaches.
* * *

---

## Author Comment (AC2) · 2 Nov 2020

**Referee comments by Anonymous Referee #1**

This study describes the main dry and humid events that have affected the Iberian Peninsula from 1901 to 2016. The research topic addressed is relevant nevertheless, the study is principally descriptive. This is not a limitation per se, but further discussion should be necessary to justify the relevance and novelty of this study in comparison to the several studies (some of them very recent) analysing droughts from different perspectives in the Iberian Peninsula.

Reply: We thank the reviewer for the constructive comments, which will hopefully contribute to improving the revised manuscript. Our detailed responses can be found below.

We thought the main objectives and motivations underpinning this manuscript were clear from the beginning. As we wrote in the abstract (lines 12-13): "Here, a method for ranking regional extremes of persistent, widespread drought and wet events is presented, considering different time scales". We also wrote in the introduction (lines 100-105):

"In summary the main goals of this paper are to:

    (i)       present a tool which allows identifying regional extremes of persistent, widespread dry and wet periods, at different time scales;

    (ii)      build a comprehensive dataset of rankings of the most extreme, prolonged, widespread drought and wet periods on Iberia (…)"

In this regard, this manuscript **does not aim at:**

1) analysing droughts from different perspectives in the Iberian Peninsula;

2) comparing or assess extremes (wet and dry) using different indices;

3) evaluating how extreme an event (wet or dry) using the impact perspective;

4) characterizing extremes from the impact perspective;

5) assessing the different mechanisms behind de extreme events;

6) assessing the climate change signal or the dependence of these extremes (wet or dry) on the different variability modes.

The authors have participated in the last 2 decades in more than 30 works that cover those different aspects of droughts in Iberia and elsewhere, but those were clearly not the focus in the current analysis. We would like simple to contribute with a simple, but yet robust methodology that is useful to rank wet and dry events, considering both their magnitude and spatial extent.

We would like to stress that **we have the purpose of ranking both wet and dry (and not to detail the case of droughts) events**. The ranking methodology is here applied to both type of episodes (wet and dry), highlighting the extreme or severe character of their occurrence.

We agree that all these are very relevant and interesting research questions, and these are some of the research questions that motivated us to identify that there is not a ranking of extreme and widespread dry and wet events, yet. However, the characterization and evaluation of impacts for both extreme (wet and dry) episodes will utterly enlarge significantly the present

manuscript. This would pose as a disadvantage in terms of the focus of the article, which we think would be much disperse.

Therefore, we will include changes that will clarify and explain better the motivation and scope of the paper. We will also highlight the relevance and novelty of this study, which is presenting a methodology for building rankings of extreme and widespread dry and wet events, for several timescales, whatever indices datasets are used (in this case we used SPEI).

In addition, there are some methodological issues that should be considered to remove some interpretations of the obtained results.

There are also some data issues and recommendations to improve the manuscript. I would suggest a major revision of this manuscript.

Reply: We thank the reviewer for these comments. A detailed response to all the reviewer's comments will follow.

**Comments:**

42. typo: Spatial extend

REPLY: The typo will be corrected.

44. I would also mention environmental or ecologic droughts (e.g. https://doi.org/10.1016/j.earscirev.2019.102953)

REPLY: We will include the reference to the recent concept of environmental or ecologic droughts, as follows (line 54):

> "*In recent years, a special focus was put in environmental or ecologic droughts when considering long and widespread dry events with strong impacts that may induce changes in natural and managed ecosystems (Crausbay et al., 2017, Vicente Serrano et al., 2020)*"

- Crausbay et al., 2017, Defining Ecological Drought for the Twenty-First Century, Bull. Amer. Meteor. Soc., 98 (12): 2543–2550. https://doi.org/10.1175/BAMS-D-16-0292.1)
- Vicente Serrano et al., 2020 A review of environmental droughts: Increased risk under global warming? Earth-Science Reviews, 201, 102953 (https://doi.org/10.1016/j.earscirev.2019.102953)

44-54. I do not think necessary here to describe in depth these drought types.

REPLY: We appreciate the reviewer's concern. However, the mentioning of the different types of droughts has the goal of highlighting the different time-scales of the dry extreme events and the need of using a multiscalar index, such as SPEI. Therefore, we opt by reducing detail, but without removing it completely.

58. The PDSI is not recent. I suggest to cite a study that reviews drought indices (e.g. https://doi.org/10.1175/1520-0477-83.8.1149, https://doi.org/10.1007/s40641-018-0098-x).

REPLY: We agree with the reviewer and the suggested comment and references will be added to the manuscript.

63-68. I would remove this criticism of the PDSI. I do not it is needed here.

REPLY: We agree with the reviewer and the manuscript will be changed accordingly.

85. This is not a water balance model. The SPEI is not intended to be soil moisture metric.

REPLY: We understand the reviewer's comment and therefore we change the sentence as follows:

"water balance is reflected on SPEI computation by the inclusion of …"

93-99. I would suggest to revise recent literature on this topic also in the Iberian Peninsula (https://doi.org/10.1002/joc.6719, https://doi.org/10.1007/s11600-018-0138-x, https://doi.org/10.1111/nyas.14365, https://doi.org/10.1002/joc.6126).

REPLY: We thank the reviewer for the suggestions and the additional literature will be included. The manuscript will be changed accordingly:

*"The characterization of wet and dry periods in the IP is extremely important as the region is frequently affected by extreme dry and wet events (González-Hidalgo et al., 2018; Domínguez-Castro et al., 2019; Vicente-Serrano et al. 2020) and consequently by their impacts on several systems (Liberato et al., 2017; Ribeiro et al., 2018, 2019) (…)"*

- Domínguez-Castro, F, Vicente-Serrano, SM, Tomás-Burguera, M, *et al.* High spatial resolution climatology of drought events for Spain: 1961–2014. *Int J Climatol*. 2019; 39: 5046– 5062. https://doi.org/10.1002/joc.6126
- González-Hidalgo, J.C., Vicente-Serrano, S.M., Peña-Angulo, D. *et al.* High-resolution spatio-temporal analyses of drought episodes in the western Mediterranean basin (Spanish mainland, Iberian Peninsula). *Acta Geophys.* **66,** 381–392 (2018). https://doi.org/10.1007/s11600-018-0138-x
- Liberato M. L. R., Ramos A. M., Gouveia C. M., Sousa P., Russo A., Trigo R.M., Santo F.E. (2017) Exceptionally extreme drought in Madeira Archipelago in 2012: Vegetation impacts and driving conditions. Agricultural and Forest Meteorology, doi: 10.1016/j.agrformet.2016.08.010
- Ribeiro, A. F. S., Russo, A., Gouveia, C. M., Páscoa, P., and Pires, C. A. L. (2019) Probabilistic modelling of the dependence between rainfed crops and drought hazard. Nat. Hazards Earth Syst. Sci., 19, 2795–2809, https://doi.org/10.5194/nhess-19-2795-2019

- Vicente-Serrano, SM, Domínguez-Castro, F, Murphy, C, *et al.* Long-term variability and trends in meteorological droughts in Western Europe (1851–2018). *Int J Climatol*. 2020; 1– 28. https://doi.org/10.1002/joc.6719

115. Nevertheless, the relevant issue is the number of stations used in the Iberian Peninsula. This is very relevant in this study. This is very relevant in this study since this may affect the obtained results since few interpolated stations may filter too much the spatial variance of the specific drought episodes.

REPLY: We acknowledge the reviewer's concern. However, we would like to point out that the CRU dataset derives from thousands of stations scattered non randomly across the Earth, with much higher densities at mid-latitudes than in the tropics or the Arctic. Although, in terms of temporal bias, the CRU dataset logically contains many fewer observations in the early part of its record, this happens mostly in the tropical and Arctic regions (Macias-Fauria et al., 2014), thus not affecting so much the more covered mid-latitude regions. Despite the smaller number of meteorological stations used to obtain the data in the first half of the 20th century when compared to the remaining years, Harris and coauthors (Harris et al., 2013) show that their precipitation and temperature time series have a high correlation value with other datasets (Harris et al., 2013). Macias-Fauria and coauthors (2014) show that for global studies the analyses should be undertaken using data after 1950s. However, the analysis of the present work is restricted to the IP region which includes a relatively homogeneous number of stations. Additional information about the number of stations for the considered variables in several years along the 20th century and also the methods used for interpolation and the representativeness of the available data is also provided in New et al. (2000) and Mitchell and Jones (2005). According to these authors the dataset is useful for a variety of climatic applications including hydrological modelling, climate change scenarios, and evaluation of regional climate models. Moreover, this database has been previously used by the authors (Russo et al., 2015; Páscoa et al., 2017), which have obtained good results in the IP, including for the earlier years (Páscoa et al., 2017).

- Harris I., P. D. Jones, T. J. Osborn, and D. H. Lister, "Updated high-resolution grids of monthly climatic observations—the CRU TS3.10 Dataset," *International Journal of Climatology*, vol. 34, no. 3, pp. 623–642, 2013.
- Macias-Fauria M., A. W. R. Seddon, D. Benz, P. R. Long, and K. Willis, "Spatiotemporal patterns of warming," *Nature Climate Change*, vol. 4, no. 10, pp. 845-846, 2014.
- Mitchell T. D. and P. D. Jones, "An improved method of constructing a database of monthly climate observations and associated high-resolution grids," *International Journal of Climatology*, vol. 25, no. 6, pp. 693–712, 2005.
- New M., M. Hulme, and P. Jones, "Representing Twentieth-Century SpaceTime Climate Variability. Part II: Development of," *Monthly Grids of Terrestrial Surface Climate. J. Climate*, vol. 13, Article ID 190196, pp. 2217–2238, 2000.
- Páscoa P., C. M. Gouveia, A. Russo, and R. M. Trigo, "The role of drought on wheat yield interannual variability in the Iberian Peninsula from 1929 to 2012," *International Journal of Biometerology*, vol. 61, no. 3, pp. 439–451, 2017.
- Russo A. C., C. M. Gouveia, R. M. Trigo, M. L. Liberato, and C. C. DaCamara, "The influence of circulation weather patterns at different spatial scales on drought variability in the Iberian Peninsula," *Frontiers in Environmental Science*, vol. 3, 2015.

Nevertheless, a sentence to highlight that the data on the first part of the XX century should be analysed with car will be added.

117. I wonder on the goodness of the VPD data for the first decades of the XXth century.

REPLY: We understand the reviewer's concern however we did not use VPD data. Comments about the quality of CRU dataset used on this work are already presented on previous comment.

132-146. There are strong uncertainties related to the calculation of the evaporative demand prior 1960s. This explains that some drought datasets based on the AED start in the decade of 1960 (https://doi.org/10.3390/data2030022). I am not convinced by the arguments provided to justify the use of the CRU dataset for the long-term, mostly for the atmospheric demand, at least in Spain few of the data necessary to calculate the AED with the FAO-56 method is not available. If author's main focus is the long term, I would recommend focusing only on precipitation data and the SPI. This would reduce uncertainty since CRU precipitation data shows good capacity to reproduce long-term precipitation variability in the region (https://doi.org/10.1088/1748-9326/ab9c4f).

REPLY: This comment is in line with the two previous. We agree with the reviewer that there are larger uncertainties present when using data prior than the 1960's. As highlighted on the previous comments, this was noted and will be highlighted in the text.

Nevertheless, we would like to point that the potential evapotranspiration used was extracted from CRU, which was calculated based on Penmann-Monteith equation. This method is considered as the standard procedure for computing PET by several international institutions such as the Food and Agriculture Organization of the United Nations (FAO), the International Commission on Irrigation and Drainage (ICID), or the American Society of Civil Engineers (ASCE).

Moreover, we again would like to stress that this database has been previously used by the authors (Russo et al., 2015; Páscoa et al., 2017), which have obtained good results in the IP, including for the earlier years (Páscoa et al., 2017).

183-185. Repeated above.

REPLY: We agree with the reviewer. The sentence will be changed in order to avoid repetitions:

*"Here, SPEI was applied with the main purpose to identify the major extensive and/or extreme droughts which affected the IP, while illustrating the relevance of the method."*

166. I suggest restricting the analysis to time scales below 12 months. The autoregressive character of SPEI and SPI affect the identification of the drought events and the possible trends (e.g., https://doi.org/10.1002/joc.6350, https://doi.org/10.1175/JCLI-D15-0590.1, https://doi.org/10.1016/j.scitotenv.2020.140094).

REPLY: We agree with the reviewer that one must be aware of the autoregressive character of SPEI and SPI. However, we would like to stress that in this work we only want to present a methodology to rank extreme wet and dry events, that is, we want to present a tool which allows

to compare extremes, taking into account both the intensity and the area affected. The reviewer is correct when highlighting that timescales higher than 12 months must be analysed cautiously. This is one of the important results of this work: by jointly comparing the rankings of extreme events (both wet and dry) among different timescales more information may be obtained on the real extremeness character of the events. Therefore, a sentence will be included in the manuscript pointing the autoregressive character of SPI and SPEI and reinforcing the need to look the results carefully.

> *"It should be noted that due to the autoregressive character of SPEI (or SPI), results obtained for larger time-scales (18 and 24 months) should be looked very carefully."*

Fig. 2 Most of the events identified are recorded in the decades of 2000 and 2005. I would ask if AED uncertainties may be biasing this issue. Droughts based on precipitation data do not show so important trends (https://doi.org/10.1002/joc.6719). Also SPEI data based on high quality data in Spain does not show so important trend (https://doi.org/10.1002/joc.6126).

REPLY: We thank the reviewer for the comment. In fact, we are aware about the impact of the AED uncertainties on trends for the recent years and we agree that we should include a sentence enriching the discussion. Furthermore, we would like to recall the reviewer that this paper is not studying droughts, nor its trends. We are presenting a methodology which allows to categorize, to rank, to hierarchize, to compare different extreme events (both dry and wet), including the affected area. Individual, recent, extreme wet/drought events are well known in the Iberian Peninsula. We can mention for example the outstanding 2004-2005 drought in the Iberian Peninsula (Garcia-Herrera et al. 2007) or the record Winter drought of 2011-12 in the Iberian Peninsula (Trigo et al. 2013). We can also mention the 2009-2010 extreme winter precipitation in the Iberian Peninsula (Vicente-Serrano et al. 2011). Therefore, severe and long drought events are frequent in the IP and are getting even more severe in the last decades (Trigo et al. 2013). This increasing severity is accompanied by an increased tendency for dryness and decrease of vegetation cover, due to the higher atmospheric evaporative demand (Vicente-Serrano et al., 2012, 2014). This is reflected on the higher trends of SPEI than for SPI in the IP.

There is no doubt that all these extreme events have been previously assessed and many questions arose. Is there a trend? Are these widespread, outstanding, extreme events becoming more intense? More frequent? These are interesting research questions, but we do not intend to reply to these questions in this paper.

However, when studying extreme events another question arises: how can we compare two outstanding events? Which (dry or wet) extreme is the most extreme? Should we consider the most extreme the longest? The largest? How should we compare a smaller, very intense extreme with a larger, less intense extreme? Therefore, this paper intends to discuss only this topic. Future work may eventually use this ranking to identify variability and trends of extremes, but this is not the scope of this paper.

> *"It should be stressed that although the high number of events (dry and wet) for the XXI century on the ranking, the present methodology does not allow (neither aims) to point to a trend of extreme events on IP during the last decades."*

- Barriopedro D., Gouveia C.M., Trigo R.M., Wang L. (2012) The 2009/10 Drought in China: Possible Causes and Impacts on Vegetation. Journal fo Hydrometeorology, Vol 13. DOI: 10.1175/JHM-D-11-074.1
- Garcia-Herrera R., Paredes D., Trigo R.M., Trigo I.F., Hernández H., Barriopedro D., Mendes M.T. (2007) The outstanding 2004-2005 drought in the Iberian Peninsula: associated atmospheric circulation. Journal ofHidrometeorology, 8, 483-498
- Sousa P., Trigo R.M., Aizpurua P., Nieto R., Gimeno L., Garcia-Herrera R. (2011) Trends and extremes of drought indices throughout the 20th century in the Mediterranean. Natural Hazards and Earth System Sciences, 11, 33-51, doi:10.5194/nhess-11-33-2011
- Trigo R.M., Añel J., Barriopedro D., García-Herrera R., Gimeno L., Nieto R., Castillo R., Allen M.R., Massey N. (2013) The record Winter drought of 2011-12 in the Iberian Peninsula, in Explaining Extreme Events of 2012 from a Climate Perspective. Bulletin of the American Meteorological Society, 94 (9), S41-S45
- Trigo R.M., Gouveia C., Barriopedro D. (2010) The intense 2007-2009 drought in the Fertile Crescent: Impacts and associated atmospheric circulation. Agricultural and Forest Meteorology, 150, 1245-1257
- Vicente-Serrano S. M., Lopez-Moreno Juan-I., Beguería S., Lorenzo-Lacruz J., Sanchez-Lorenzo A., García-Ruiz J. M., Azorin-Molina C., Morán-Tejeda E., Revuelto J., Trigo R., Coelho F., Espejo F. (2014) Evidence of increasing drought severity caused by temperature rise in southern Europe. Environmental Research Letters, doi:10.1088/1748-9326/9/4/044001
- Vicente-Serrano S. M., S. Beguería, J. Lorenzo-Lacruz et al., (2012) "Performance of drought indices for ecological, agricultural, and hydrological applications," *Earth Interactions*, vol. 16, no. 10, pp. 1–27.
- Vicente-Serrano S.M., Trigo, R.M., Liberato M.L.R., López-Moreno J.I., Lorenzo-Lacruz J., Beguería S., Morán-Tejeda H., El Kenawy A. (2011) Extreme winter precipitation in the Iberian Peninsula, 2010: anomalies, driving mechanisms and future projections. Climate Research , 46, 51-65, doi: 10.3354/cr00977

221-234. Remove 18 and 24 months. Persistence should be also recorded with 6-and 12-month SPEI and it is not affected by the mentioned autocorrelation problems.

REPLY: Please see previous answer to your comment on line 166.

236-244. I would not use different thresholds. This makes the structure of the article more complex unnecessarily.

REPLY: As we have explained previously, we are hierarchizing extreme events taking into account both the intensity and the affected area. This is the novelty. The methodology aims at assessing how intensity and area contribute to the extreme events. Therefore, a sensitivity analysis on the impact of different thresholds to define dry and wet events is performed. This is fundamental to assess how the ranking depends on the intensity.

Figure 1. Is this needed?

REPLY: We agree that for someone who studies the Iberian Peninsula this figure would not be relevant. However, the scope of this journal is global, and the figure helps someone who is not so familiar with Europe to better identify the study region.

Discussion: There is not a discussion section that discusses in depth the obtained results and some comparison with the scientific literature related to droughts published in the Iberian Peninsula in recent years. I think extremely necessary to write an independent discussion section in which the caveats of the data and methodology are stressed, but also the implications in the current climate change scenario and the possible novelty of the results in relation with the existing literature.

REPLY: We agree with the reviewer that a discussion section may be included to stress some of the identified caveats, advantages and disadvantages of this methodology. Nevertheless, we would like to stress that, to the best of our knowledge, this is the first and only paper to present a methodology to rank extreme (dry and wet) events in the Iberian Peninsula. Moreover, we stress, this paper does not study droughts (or floods). However, recognizing the need for a more proficient discussion, as pointed by the reviewer's comments, we will include in the discussion section some highlights for future research questions which may be addressed after a ranking methodology and dataset is published.

Thank you!

---

## Author Comment (AC3) · 2 Nov 2020

**Referee comments by Anonymous Referee #2**

The main goal of this study is to characterize and rank the most extreme, widespread drought (wet) events in the Iberian Peninsula between 1901–2016. The method used is based on the multiscalar Standardized Precipitation Evapotranspiration Index (SPEI) from CRU. Besides, different time scales (6, 12, 18 and 24 months) were considered in ranking. The topic is quite interesting, but there is a lack of further discussion of the results. It is also necessary to include some validation of the most severe drought events identified in the study area.

Reply: We acknowledge the reviewer for the support and constructive comments, which may contribute to improving the revised manuscript. We agree with the reviewer that a discussion section may be included to stress some of the identified caveats, advantages and disadvantages of this methodology. We will also develop the analysis by including some comparison with auxiliary information on the most severe (wet and dry) events identified in the study area. Our detailed responses can be found below.

**General Comments:**

Abstract: The authors could bring more highlights about the results in the abstract.

REPLY: We agree with the reviewer and the abstract will be improved accordingly.

Introduction:

Line 57: It is not correct to say that drought indices have been developed in recent years. The PDSI, for example, was created more than 50 years ago. Please modify the sentence.

REPLY: We agree with the reviewer and the manuscript will be changed accordingly.

Lines 106-109: Perhaps these sentences can be removed. I do not think it is necessary to describe the topics of the article.

REPLY: We agree with the reviewer, these sentences can be removed.

The authors could add more details in the introduction about the reason for using SPEI instead other indices, pointing out the main advantages and disadvantages.

REPLY: We agree with the reviewer and we will include more on this information, namely mentioning works performed with SPEI and other indices on IP (Gouveia et al., 2017, Pascoa et al. 2017a, 2017b, Ribeiro et al., 2018). However, we would like to highlight that for this manuscript we could either apply SPEI or other indices; this work does not aim at comparing or assess extremes (wet and dry) using different indices. This is a relevant topic, but it is not the scope here.

*"Several authors have performed drought characterization and assessment of impacts, namely on vegetation dynamics (Gouveia et al., 2016, Liberato et al., 2018) and on crops production and yield (Pascoa et al. 2017a, 2017b, Ribeiro et al., 2018). The SPEI and SPI were used in order to identify drought severity and intensity and both indices are able to identify the severest and longest events. Although the recent issues raised due to the uncertainties on AED computation (https://doi.org/10.1002/joc.6719, https://doi.org/10.1002/joc.6126), some works highlighted the ability of SPEI to identify tendencies towards to dryer conditions on Iberian Peninsula (Vicente-serrano et al., 2014, Spinoni et al., 2015, Coll et al., 2016, Páscoa et al. 2017)."*

- Coll J., E. Aguilar, and L. Ashcroft, "Drought variability and change across the Iberian Peninsula," Theoretical and Applied Climatology, vol. 130, no. 3-4, pp. 901–916, 2016.
- Páscoa P, Gouveia CM, Russo A, Trigo RM (2017a) Drought Trends in the Iberian Peninsula over the Last 112 Years. Advances in Meteorology Volume 2017, Article ID 4653126, 13 pages https://doi.org/10.1155/2017/4653126
- Páscoa P., C. M. Gouveia, A. Russo, R. M. Trigo (2017b) The role of drought on wheat yield interannual variability in the Iberian Peninsula from 1929 to 2012. Int J Biometeorol, 61:439-451 DOI: 10.1007/s00484-016-1224-x
- Ribeiro, A. F. S., Russo, A., Gouveia, C.M., Páscoa, P. (2018). Modelling drought-related yield losses in Iberia using remote sensing and multiscalar indices, Theor Appl Climatol, doi:10.1007/s00704-018-2478-5
- Spinoni J., G. Naumann, J. Vogt, and P. Barbosa, "European drought climatologies and trends based on a multi-indicator approach," Global and Planetary Change, vol. 127, pp. 50–57, 2015.
- Vicente-Serrano S. M., J.-I. Lopez-Moreno, S. Begueria et al., "Evidence of increasing drought severity caused by temperature rise in southern Europe," Environmental

Extensive extreme droughts during the period 1901-2016: How the major negative societal impacts related to severe droughts was measured? Please, add some examples in the manuscript.

REPLY: We agree with the reviewer that major negative societal impacts are one of the motivations for better understanding extreme, widespread events (dry or wet). As stated above we will include additional information on the most severe (wet and dry) events identified in the study area, including some information on negative societal impacts.

However, we would like to stress here that this paper is not studying the impacts of droughts. We are mostly presenting a methodology which allows to categorize, to rank, to hierarchize, to compare different extreme events (both dry and wet), including the affected area and the magnitude. Individual, recent, extreme wet/drought events are well known in the Iberian Peninsula and have been studied by some of the authors in recent years. We can mention for example the outstanding 2004-2005 drought in the Iberian Peninsula (Garcia-Herrera et al. 2007) or the record Winter drought of 2011-12 in the Iberian Peninsula (Trigo et al. 2013). We can also mention the 2009-2010 extreme winter precipitation in the Iberian Peninsula (Vicente-Serrano et al. 2011). All these extreme events have been previously studied. To develop a methodology to assess these extreme events through the impact perspective is another interesting research question, but it is out of the scope of this paper.

Please consider including some information on the possible causes (climate) of the most severe drought events in the study area.

REPLY: We thought the main objectives and motivations underpinning this manuscript were clear from the beginning. From the reviewers' comments we verify that the scope of this manuscript is not clearly stated. Therefore, we will include changes that will clarify and explain better the motivation and objectives of the paper. We will also highlight the relevance and novelty of this study, which is presenting a methodology for building rankings of extreme and widespread dry and wet events, for several timescales, whatever indices datasets are used (in this case we used SPEI).

To accommodate the reviewer's concerns, we will also include in the discussion a review on the possible causes (including climate) associated with the occurrence of the most severe drought and wet events in the Iberian Peninsula. However, highlighting that attribution of few extreme events is a hot research topic which is out of the scope here.

In addition, it is necessary to quantify the droughts impacts on agricultural production and reservoirs, as a validation of the ranking of drought.

REPLY: We understand the reviewer's point. In fact, several works have already presented the impacts of droughts on vegetation dynamics and crops production and yields (Gouveia et al., 2009, 2012; Páscoa et al., 2017; Ribeiro et al., 2018, 2019) over Iberia. However, we must stress that the assessment of the impacts of dry and wet events is not the scope of this manuscript. This can be seen as a caveat from the point of view of risk management. Therefore, we will be pointing to the ranked impacts of wet and dry episodes and highlight that the impacts were not included in the construction of the methodology.

Figure 1 - What information is presented by the color scale in figure 1? Please specify if it is a physical variable (elevation, etc.)

REPLY: Thank you for highlighting this. We will include a color scale, elevation and units.

Thank you!

---

## Author Comment (AC4) · 2 Nov 2020

Reply: We acknowledge the reviewer for the support and constructive comments, which will definitely contribute to improving the revised manuscript. We agree with the reviewer that a more in depth discussion may be provided, including the topics mentioned by the reviewer below. We also agree that we must emphasize why developing such a ranking is fundamental for future research on (dry and wet) extremes – either from the impact perspective, for understanding the physical mechanisms behind each of these events, for the attribution to climate (natural or forced) variability or to climate change, for the development of future risk assessments. . . In this regard

the authors would like to stress that a lot of publications have been published in the last two decades covering these topics, including many co-authored by some of us in terms of physical mechanism (e.g. Garcia-Herrera et al. 2007, Trigo et al. 2013, Vicente-Serrano et al. 2011) or in terms of impacts (e.g. Gouveia et al., 2009, 2012 Páscoa et al., 2017, Andreia et al., 2018. In any case, after this ranking is established, additional investigation on these research topics may be performed. However, we would like to stress that the scope of this paper is, as stated in the introduction (lines 100-105): "In summary the main goals of this paper are to: (i) present a tool which allows identifying regional extremes of persistent, widespread dry and wet periods, at different time scales; (ii) build a comprehensive dataset of rankings of the most extreme, prolonged, widespread drought and wet periods on Iberia (. . .)" Our detailed responses can be found in the attached file.

Please also note the supplement to this comment:
https://esd.copernicus.org/preprints/esd-2020-46/esd-2020-46-AC4-supplement.pdf

**Supplement:**

**Referee comments by Anonymous Referee #3**

**Main comments**: The manuscript presents a new method for ranking regional extremes of persistent, widespread drought and wet events that considers different time scales. While there are some good points in the manuscript, it lacks in discussion where the authors could put things in context.

Reply: We acknowledge the reviewer for the support and constructive comments, which will definitively contribute to improving the revised manuscript. We agree with the reviewer that a more in depth discussion may be provided, including the topics mentioned by the reviewer below. We also agree that we must emphasize why developing such a ranking is fundamental for future research on (dry and wet) extremes – either from the impact perspective, for understanding the physical mechanisms behind each of these events, for the attribution to climate (natural or forced) variability or to climate change, for the development of future risk assessments… In this regard the authors would like to stress that a lot of publications have been published in the last two decades covering these topics, including many co-authored by some of us in terms of physical mechanism (e.g. Garcia-Herrera et al. 2007, Trigo et al. 2013, Vicente-Serrano et al. 2011) or in terms of impacts (e.g. Gouveia et al., 2009, 2012 Páscoa et al., 2017, Andreia et al., 2018. In any case, after this ranking is established, additional investigation on these research topics may be performed.

However, we would like to stress that the scope of this paper is, as stated in the introduction (lines 100-105):

"In summary the main goals of this paper are to:

(i)     present a tool which allows identifying regional extremes of persistent, widespread dry and wet periods, at different time scales;
(ii)    build a comprehensive dataset of rankings of the most extreme, prolonged, widespread drought and wet periods on Iberia (…)"

**Specific questions** that need to be addressed are:

1.

Can you show that this method using SPEI performs better for the Iberian Peninsula?

How does it compare with other indices (say SPI) for the same purpose?

REPLY: We understand the reviewer's curiosity. In light of our results and the sensitivity analysis performed on the impact of different thresholds to define dry and wet events, using different indices will certainly change the positions of the events, as it is highlighted (lines 158-162) when analysing rankings at different timescales (and for the same index). Once again, we do not intend to show that this method performs better using one or another index for the Iberian Peninsula or other region – we will stress this point in the discussion section.

Once the methodology is established, one of the interesting studies that may be performed is, precisely, to study how the ranking's positions compare when using other indices (say SPI). This analysis could open new perspectives to the understanding the dynamics behind the of the most

extreme and widespread dry and wet events in the Iberian Peninsula. Eventually the behaviour is different between wet and dry extremes? By applying to different regions and indices the research on the most extreme events may be developed.

However, we would like to highlight that for this manuscript we could either apply SPEI or other indices; this work does not aim at comparing or assess extremes (wet and dry) using different indices. This is a relevant topic, but it is not the scope here. We will include more on this information, namely mentioning works performed with SPEI and other indices on IP (Gouveia et al., 2017, Pascoa et al. 2017a, 2017b, Ribeiro et al., 2018).

Can you validate its applicability for the outcomes (hydrological/ ecological/ agricultural) on different time-scales?

REPLY: Please see the answer to previous comments.

It should be fairly easy to show like Vicente-Serrano et al., 2012 did (for global and continental scales) that SPEI is better than SPI (and PDSI) for various applications. I wonder if it is true for the Iberian Peninsula also?

REPLY: Some of this manuscript co-authors have already performed a set of works comparing SPEI and SPI results for Mediterranean and PI in particular (Gouveia et al. 2017, Pascoa et al., 2018, among others). We will include this information in the manuscript:

> *"Several authors have performed drought characterization and assessment of impacts, namely on vegetation dynamics (Gouveia et al., 2016, Liberato et al., 2018) and on crops production and yield (Pascoa et al. 2017a, 2017b, Ribeiro et al., 2018). The SPEI and SPI were used in order to identify drought severity and intensity and both indices are able to identify the severest and longest events. Although the recent issues raised due to the uncertainties on AED computation (Domínguez-Castro et al., 2019; Vicente-Serrano et al., 2020), some works highlighted the ability of SPEI to identify tendencies towards to dryer conditions on Iberian Peninsula (Vicente-serrano et al., 2014, Spinoni et al., 2015, Coll et al., 2016, Páscoa et al. 2017)."*

- Domínguez-Castro, F, Vicente-Serrano, SM, Tomás-Burguera, M, *et al.* High spatial resolution climatology of drought events for Spain: 1961–2014. *Int J Climatol*. 2019; 39: 5046– 5062. https://doi.org/10.1002/joc.6126
- Coll J., E. Aguilar, and L. Ashcroft, "Drought variability and change across the Iberian Peninsula," Theoretical and Applied Climatology, vol. 130, no. 3-4, pp. 901–916, 2016.
- Páscoa P, Gouveia CM, Russo A, Trigo RM (2017a) Drought Trends in the Iberian Peninsula over the Last 112 Years. Advances in Meteorology Volume 2017, Article ID 4653126, 13 pages https://doi.org/10.1155/2017/4653126
- Páscoa P., C. M. Gouveia, A. Russo, R. M. Trigo (2017b) The role of drought on wheat yield interannual variability in the Iberian Peninsula from 1929 to 2012. Int J Biometeorol, 61:439-451 DOI: 10.1007/s00484-016-1224-x

- Ribeiro, A. F. S., Russo, A., Gouveia, C.M., Páscoa, P. (2018). Modelling drought-related yield losses in Iberia using remote sensing and multiscalar indices, Theor Appl Climatol, doi:10.1007/s00704-018-2478-5
- Spinoni J., G. Naumann, J. Vogt, and P. Barbosa, "European drought climatologies and trends based on a multi-indicator approach," Global and Planetary Change, vol. 127, pp. 50–57, 2015.
- Vicente-Serrano S. M., J.-I. Lopez-Moreno, S. Begueria et al., "Evidence of increasing drought severity caused by temperature rise in southern Europe," Environmental
- Vicente-Serrano, SM, Domínguez-Castro, F, Murphy, C, *et al.* (2020) "Long-term variability and trends in meteorological droughts in Western Europe (1851–2018)". *Int J Climatol*.; 1– 28. https://doi.org/10.1002/joc.6719

2.

The question of ranking also needs more exploration. Does a top ranking translate into maximum impacts on the indicators (agriculture/ streamflow etc.) for the relevant time-scale?

Can this be shown in an analysis?

REPLY: We acknowledge the reviewer for identifying the importance of establishing a methodology for building a ranking on the extreme and widespread dry and wet events in the Iberian Peninsula (which may be replicated to any other study region). We agree that this and all the questions raised below are very relevant and interesting research questions, and these are some of the research questions that motivated us to identify that there is not a ranking of extreme and widespread dry and wet events, yet. And only after this ranking is established will we be able to perform research on all other research topics mentioned by the reviewer.

Therefore, we will change the manuscript and explain better the motivation and scope of the paper. We will also justify better the relevance and novelty of this study, which is presenting a methodology for building rankings of extreme and widespread dry and wet events, for several timescales, whatever indices datasets are used (in this case we used SPEI).

Additionally, we will include a discussion section to stress some of the identified caveats, advantages and disadvantages of this methodology. Finally, we will refer some highlights, topics that need more exploration, research questions which may be addressed in the future after a ranking dataset is published.

> *"The years ranked in the first places either for the drought or wet evens have being object of study in several previous works (Gouveia et., 2009, 2012, Garcia-Herrera et al., 2007, Trigo et al., 2013, Sousa et al., 2011), namely concerning with their impacts on vegetation dynamics (Gouveia et al., 2009, Liberato et al., 2017, Gouveia et al., 2016, 2017) and on crop productions and yield (Pascoa et al., 2017b, Ribeiro et al., 2019)."*

3.

The temporal variation in the indices (especially in Figures 3 & 4) described as "There is a clear temporal clustering of most extreme drought episodes, particularly with a large concentration between 1943 and 1957 and a second group after 1975".

REPLY: We agree with the reviewer. We must improve the discussion of these results and some literature review may be added to give some clues on this clustering of drought events towards the end of the data period.

> *"Páscoa and co-authors (2017) evaluate the long-term evolution of drought in the Iberian Peninsula using both SPEI and SPI at 12-month time scales computed using data from the Climatic Research Unit (CRU) database for the period of 1901–2012 and for three subperiods: 1901–1937, 1938–1974, and 1975–2012. Both indicators agree in identifying the most intense and long dry episodes during the 1940, 1950 and after the 1980 decades. They also identify the 1960s and 1970s as predominantly wet whereas the 1910s and 1930s showed shorter wet and dry periods (Vicente-Serrano et al., 2006). The recent 2004/2005 and 2011/2012 extreme drought events are also clearly evident on all indices used (Páscoa et al., 2017). These were also highlighted in the works by González-Hidalgo et al. (2018), Domínguez-Castro et al. (2019) and Vicente-Serrano et al. (2020)."*

4.Some analysis should be done (or connections to existing studies made) to analyze the role of modes of variability on decadal or multidecadal or inter-annual (such as the NAO by Vicente-Serrano et al. (2011) time-scales.

REPLY: We agree, again, with the reviewer. This is another important research question to develop in a future work. In fact some preliminary studies related with the relationship between tele connexions and drought/wet events have be already published by some of the co-authors of the present study (Gouveia et al., 2008, Sousa et al., 2011; Bastos et al., 2016).

Moreover, we will include a small paragraph with literature review and discussion in the revised version of the manuscript.

- Bastos A, Janssens IA, Gouveia CM, Trigo RM, Ciais P, Chevalier F, Peñuelas J, Rodenbeck C, Piao S, Friedlingsein P, Running SW (2016) European CO2 sink influenced by NAO and East-Atlantic Pattern coupling, Nature Communications, 7, 10315. DOI: 10.1038/NCOMMS10315
- Gouveia C., Trigo R.M., DaCamara C.C., Libonati R., Pereira J.M.C. (2008) The North Atlantic Oscillation and European vegetation dynamics. International Journal of Climatology, DOI: 10.1002/joc.1682.
- Sousa PM, Trigo RM, Aizpurua P, Nieto R, Gimeno L, Garcia-Herrera R, (2011) Trends and extremes of drought indices throughout the 20th century in the Mediterranean. Nat. Hazards Earth Syst. Sci. 11: 33–51, doi: 10.5194/nhess-11-33-2011.

Are there any studies carried out over a larger region - like say the Mediterranean – that one can make a connection to?

REPLY:

To the best of our knowledge, this is the first and only paper to present a methodology to rank extreme, widespread (dry and wet) events. This is the novelty. There are many studies carried out for larger regions on droughts, on precipitation events at daily scales, using meteorological or gridded datasets covering the entire Mediterranean basin (e.g., Sousa et al., 2011, Gouveia

et al., 2017) or more single sectors (Gouveia et al., 2009, 2012, 2016, Trigo et al., 2011, Páscoa et al. 2020). However, to the best of our knowledge there are no other widespread, extreme rankings than those published by our group (please see Ramos et al. 2014; 2017 and references therein) and these are mostly focused on precipitation. Moreover, it should be stressed that the main drought or wet events described in the papers mentioned (and other) correspond to the years ranked in the first positions.

- Gouveia CM, Bistnas I, Liberato MLR, Bastos A, Koutsias N, Trigo RM (2016) The outstanding synergy between drought, heatwaves and fuel on the 2007 Southern Greece exceptional fire season, Agricultural and Forest Meteorology, 218-219, 135-145. DOI: 10.1016/j.agrformet.2015.11.023
- Gouveia CM, Trigo RM, Berguería S, Vicente-Serrano SM (2017) Drought impact on vegetation activity in the Mediterranean region: an assessment using remote sensing data and multi-scale drought indicators, Global and Planetary Change, 151, 15-27. DOI: 10.1016/j.gloplacha.2016.06.011
- Páscoa P, Gouveia CM, Russo AC, Bojariu R, Vicente-Serrano SM, Trigo RM (2020) Drought impacts on vegetation in southeastern Europe, Remote Sensing, 12 (13), 2156. DOI: 10.3390/rs12132156
- Ramos AM, Trigo RM, and Liberato MLR. (2014) A ranking of high-resolution daily precipitation 381 extreme events for the Iberian Peninsula. Atmospheric Science Letters 15: 328–334. DOI: 382 10.1002/asl2.507, 2014
- Ramos AM, Trigo RM, Liberato MLR (2017) Ranking of multi-day extreme precipitation events over the Iberian Peninsula, International Journal of Climatology, 37 (2), 607-620. DOI: 10.1002/joc.4726
- Trigo RM, Gouveia CM, Barriopedro D (2010) The intense 2007-2009 drought in the Fertile Crescent: Impacts and associated atmospheric circulation, AGRICULTURAL AND FOREST METEOROLOGY, 150 (9), 1245-1257. DOI: 10.1016/j.agrformet.2010.05.006

5.

Is the clustering of drought events towards the end of the data period in any way connected to climate change? There are numerous studies that have already documented drying of the Mediterranean under climate change.

REPLY: We cannot easily reply to this relevant question. In fact, some of the co-authors are preparing a manuscript about the subject that will be submitted soon. Moreover, from the reviewers' comments we verify that the scope of this manuscript is not clearly stated.

This manuscript does not aim at characterizing droughts in the Iberian Peninsula; it does not aim at assessing the climate change signal or the dependence and/or attribution of these extremes (wet or dry) on climate change or on the different variability modes.

We agree that all these are very relevant and interesting research questions, and these are some of the research questions that motivated us to identify that there is not a ranking of extreme and widespread dry and wet events, yet. And only after this ranking is established will we be able to perform more research on all other research topics mentioned above.

As the reviewer mentions, there are numerous studies that have already documented drying of the Mediterranean under climate change; there are also many studies for the Iberian Peninsula,

which climate is different. This is therefore an important research topic which is out of the scope of this paper.

However, we agree that we should improve the discussion of these results and some literature review may be added to this clustering of drought events towards the end of the data period (as previously stated on reply to point 3).

Other suggestions:

Numerous grammatical mistakes that can be easily fixed in a modern word processor.

Many thank you for your positive feedback!